# Differential modulation of positive and negative prediction errors by stimulus variability in the mouse posterior parietal cortex

Benjamin Leonardon [1,4], Sergej Kasavica[1,4], Constanze Raltschev[1], Walter Senn[1], Katharina A. Wilmes [1,2,5] & Shankar Sachidhanandam [1,3,5]

To function in uncertain environments, we need to constantly update our internal model of the world. Sensory prediction errors are thought to drive these updates. EEG studies indicate that uncertainty affects neural correlates of prediction errors. It is unclear how uncertainty of the environment modulates prediction error responses in individual neurons. Here, we exposed awake head-restrained mice to sounds followed by tactile stimuli that were either fixed or variable in intensity, making them more or less predictable. Using tactile stimuli that were stronger or weaker than expected, we identified positive and negative prediction error neurons in layer 2/3 of the posterior parietal cortex with 2-photon calcium imaging. We show that positive prediction errors are upregulated while negative prediction errors are downregulated by variability. Finally, through modelling we show that variability-dependent modulation of inhibitory neurons best fits the data, suggesting a previously unrecognized role for inhibition in encoding uncertainty prediction errors.

Learning to associate sensory stimuli through experience is a fundamental cognitive process. When the relationship between stimuli is clear and concise, we can make strong assumptions and reliably predict the stimuli. On the other hand, when the relationship is more variable, we are less able to predict the stimuli precisely. The predictive processing framework postulates that the brain builds an internal model of the external world through experience and constantly compares this to external sensory stimuli[1,2]. Sensory prediction errors are generated when there is a mismatch between what was expected and the external sensory input and are subsequently used to update the internal model[3]. Sensory prediction errors, distinct from reward prediction errors, can be categorized into positive (stimuli that are larger than expected) and negative (stimuli that are smaller than expected) deviations from the prediction[4]. These two types of sensory prediction errors were postulated to be represented by non-overlapping classes of neurons, termed positive and negative prediction error neurons[4]. Such neurons have been reported in the primary visual cortex (V1)[5], and in the posterior parietal cortex (PPC)[6,7] in mice. Their mismatch responses scale with the size of the deviation[8,9]. Whether and how these sensory prediction error neurons

also encode the variability of stimuli within learned sensory associations is unknown so far.

Here, we addressed the encoding of sensory prediction errors in an uncertain environment. We passively presented awake head-restrained mice with a sensory sequence where sound predicted a consecutive tactile stimulus with variable intensities. By associating different sounds with whisker stimuli of differing variabilities, we identified positive and negative prediction error neurons in layer 2/3 of the PPC using in-vivo 2-photon calcium imaging. We demonstrate that the PPC can report both positive and negative prediction errors that scale with the size of the deviation from the learned association. We further found that both positive and negative prediction errors were modulated by variability. Unexpectedly, however, positive and negative prediction errors were modulated in opposite ways: while the activity of positive prediction errors increased with variability, the activity of negative prediction errors decreased. We analyzed four different models which could explain the differential change in positive and negative prediction errors. Each of them provides distinct experimental predictions. A quantitative fit revealed

[1]Department of Physiology, University of Bern, Bern, Switzerland. [2]Present address: Institute of Neuroinformatics, ETH Zurich, Zurich, Switzerland. [3]Present address: Laboratory of Sensory processing, Faculty of Life Sciences, Brain Mind Institute, École Polytechnique Fédérale de Lausanne (EPFL), Lausanne, Switzerland. [4]These authors contributed equally: Benjamin Leonardon, Sergej Kasavica. [5]These authors jointly supervised this work: Katharina A. Wilmes and Shankar Sachidhanandam. ✉e-mail: katharina.wilmes@uzh.ch; shankar.sachidhanandam@unibe.ch

that a model, in which inhibitory interneurons are involved in the modulation, best replicated the data.

## RESULTS

### Sensory prediction errors scale with the size of the deviation at the PPC

We presented mice with an auditory tone followed immediately by a whisker stimulus, establishing a consistent audio-tactile sequence where the sound served as a predictor to the upcoming whisker stimulus. This protocol allowed us to create positive and negative mismatches in the audio-tactile sequence by increasing or decreasing the whisker stimulus intensity, enabling us to subsequently study how sensory prediction errors can represent deviations from learned associations. We then asked if sensory prediction errors in the PPC scale with the size of the deviation from the prediction. To test this, we paired a looming sound (sound A) with a whisker stimulus intensity of 30% and a non-looming sound (sound B) with a whisker stimulus intensity of 50%. Both audio-tactile sequences were randomly presented to awake head-fixed mice ("pairing" session of 160–200 trials, Fig. 1a). This was followed by an "interleaved" session 1 (200 trials), where we introduced a positive mismatch in the experienced audio-tactile sequences by randomly increasing the whisker stimulus intensity paired

with sound A to 90% in 15% of the trials. Hence interleaved session 1 comprised matched trials (sound A followed by whisker stimulus 30% intensity, sound B followed by whisker stimulus 50% intensity) and mismatch trials (sound A followed by whisker stimulus 90% intensity in 15% of trials, Fig. 1a). This was followed by interleaved session 2 (200 trials), where the positive mismatch was presented in the audio-tactile sequence with sound B (whisker stimulus intensity increased from 50% to 90%, in 15% of trials). We measured neuronal activity within the PPC using 2-photon calcium imaging at single-cell resolution in layer 2/3 of Thy-1GCaMP6f mice expressing the genetically encoded calcium indicator (GECI) GCaMP6f in excitatory neurons across the neocortex[10], with a chronic cranial window in place. We used intrinsic optical signal (IOS) imaging (see Methods) to localize the PPC via exclusion, where mismatch responses to the omission of tactile stimuli were previously reported[6,7]. Here, we present the mean $\Delta F/F$ computed during a 1 s window from stimulus onset minus the baseline mean $\Delta F/F$ 1 s prior to stimulus. In interleaved session 1, we identified 63 neurons ($n = 5$ mice, imaged across 8 sessions, see Methods) that responded to the positive mismatch after sound A, where the stimulus deviation was $\Delta 60\%$ (from 30% to 90%). A comparable number of neurons ($n = 56$ neurons) responded to the positive mismatch in interleaved session 2, where the stimulus deviation was $\Delta 40\%$ (from 50% to 90%). The positive

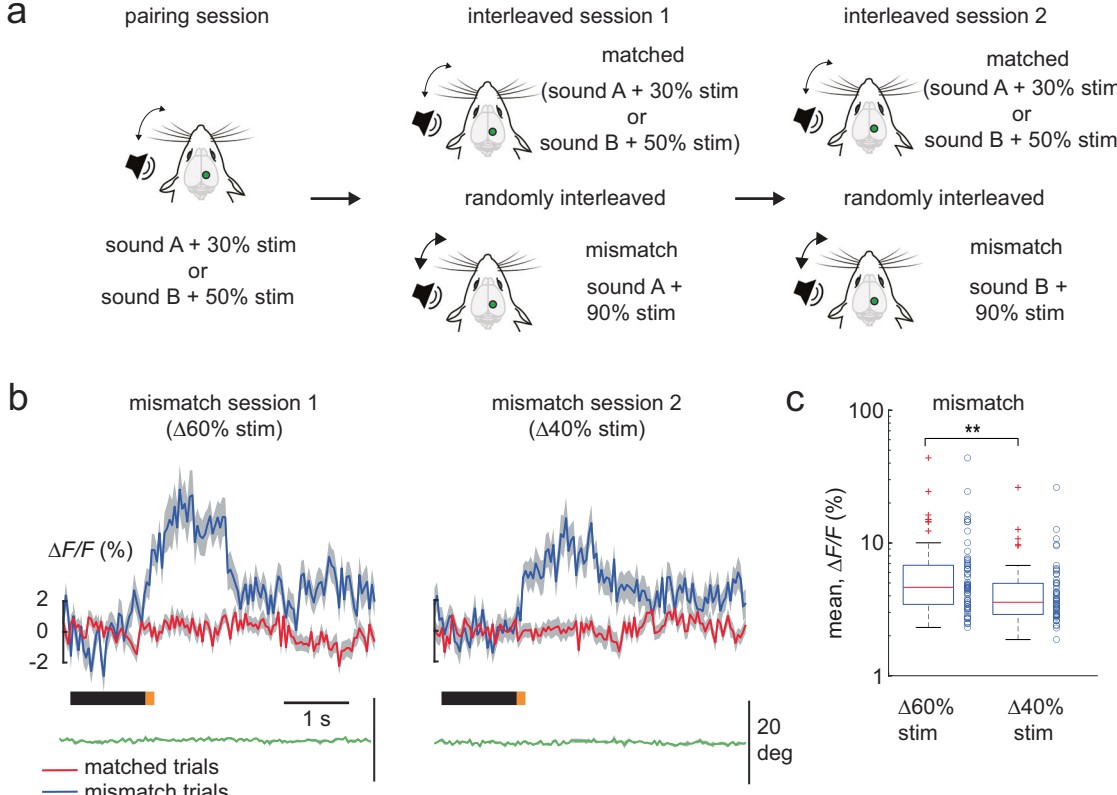

**Fig. 1 | Positive prediction errors scale with whisker stimulus deviation.**
**a** Experimental design to evaluate the scaling of positive prediction errors in a single imaging session. During the pairing session, sound A is paired with a whisker stimulus intensity of 30%, and sound B with 50% whisker stimulus intensity. Interleaved session 1 and 2 presented positive mismatch trials where the whisker stimulus intensity was randomly increased to 90% in 15% of trials. Therefore, interleaved sessions included randomized matched (sound A with 30% whisker stimulus intensity, sound B with 50% whisker stimulus intensity) and mismatch trials (session 1: sound A with 90% whisker intensity; session 2: sound B with 90% whisker intensity). **b** Population averages of $\Delta F/F$ traces of mismatch-responsive neurons in session 1 ($\Delta 60\%$ whisker stimulus intensity, $n = 63$ mismatch neurons of 703 neurons) and session 2 ($\Delta 40\%$ whisker stimulus intensity, $n = 56$ mismatch neurons of 703 neurons), with their corresponding matched (red) and mismatch

(blue) trial averages (5 Thy-1GCaMP6f mice; 8 FOV). The average whisker position from the above mismatch trials is shown below in green here and in the subsequent figures. The black and orange bars represent the sound and whisker stimuli, respectively, here and in all the subsequent figures. **c** Box plot of average population responses of mismatch-responsive neurons in (**b**). mean response $\Delta F/F$: $\Delta 60\%$ whisker stimulus intensity 6.6 ± 0.8%, $\Delta 40\%$ whisker stimulus intensity 4.7 ± 0.5%. Data are represented as mean ± s.e.m. Boxplot central line indicates the median, the bottom and top edges of the box indicate the 25th and 75th percentiles respectively, the whiskers extend to maximum and minimum points within 1.5 s.d., and outliers are marked with crosses. Statistical significance is indicated by ** for $p < 0.01$ with two-sided Wilcoxon-Mann-Whitney test. The mouse illustrations in panel **a** are adapted from[6].

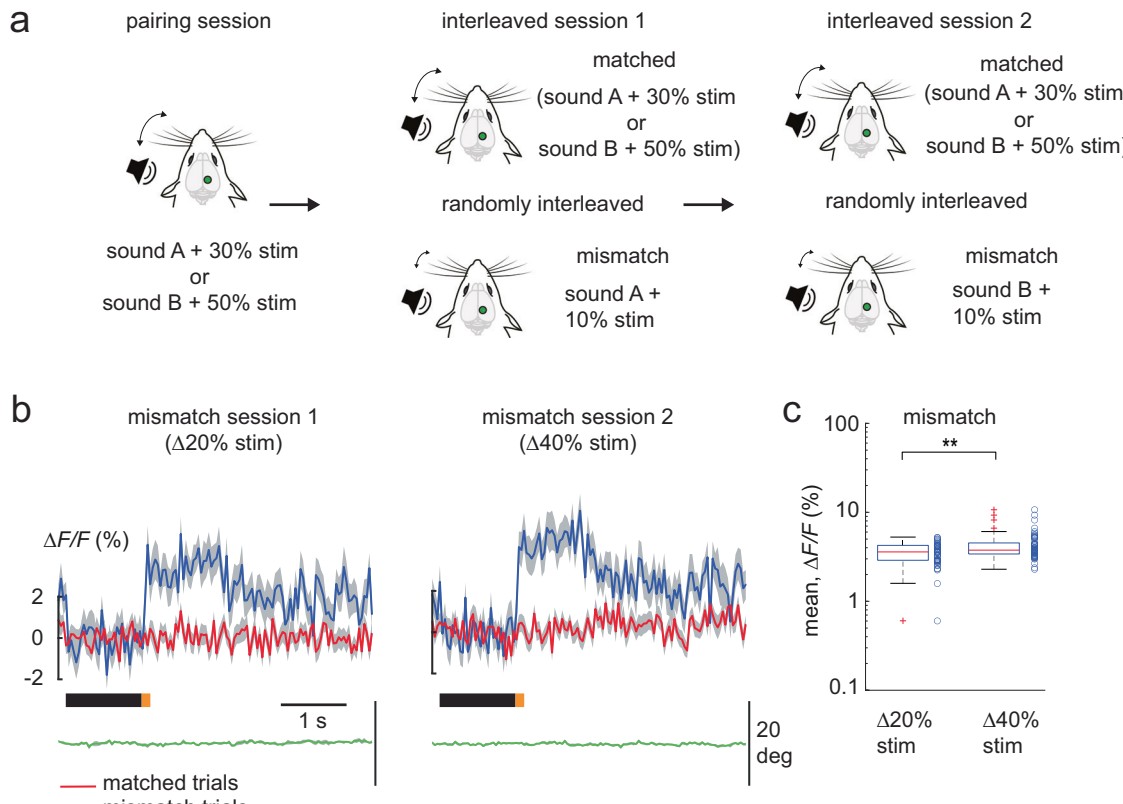

**Fig. 2 | Negative prediction errors scale with whisker stimulus deviation.**
**a** Experimental design to evaluate the scaling of negative prediction errors in a single imaging session. Pairing session is identical to that in Fig. 1. Interleaved session 1 and 2 presented negative mismatch trials by randomly decreasing the whisker stimulus intensity to 10% in 15% of trials. Interleaved sessions hence included randomized matched (sound A with 30% whisker stimulus intensity; sound B with 50% whisker stimulus intensity) and mismatch trials (session 1: sound A with 10% whisker stimulus intensity; session 2: sound B with 10% whisker stimulus intensity).
**b** Population averages of ΔF/F traces of mismatch-responsive neurons in session 1 (Δ20% whisker stimulus intensity, $n = 55$ mismatch neurons of 577 neurons) and session 2 (Δ40% whisker stimulus intensity, $n = 61$ mismatch neurons of 577

neurons), with their corresponding matched (red) and mismatch (blue) trial averages (5 Thy-1GCaMP6f mice; 7 FOV). **c** Box plot of average population responses of mismatch-responsive neurons in (**b**). mean response ΔF/F: Δ20% whisker stimulus intensity 3.5 ± 0.1%, Δ40% whisker stimulus intensity 4.2 ± 0.2%. Data are represented as mean ± s.e.m. Boxplot central line indicates the median, the bottom and top edges of the box indicate the 25th and 75th percentiles respectively, the whiskers extend to maximum and minimum points within 1.5 s.d., and outliers are marked with crosses. Statistical significance is indicated by ** for $p < 0.01$ with two-sided Wilcoxon-Mann-Whitney test. The mouse illustrations in (**a**) are adapted from ref. 6.

mismatch response increased with the stimulus deviation, being larger for the Δ60% as compared to the Δ40% in stimulus intensity (Fig. 1b, c). Hence, in the PPC, positive prediction errors increase with the size of the deviation from the learned association.

We then asked if negative prediction errors in the PPC change in a similar fashion. In a separate imaging session, we again paired the looming sound (sound A) with a whisker stimulus intensity of 30% and the non-looming sound (sound B) with a whisker stimulus intensity of 50%. As in Fig. 1, both audio-tactile sequences were presented in a random fashion ("pairing" session of 160 to 200 trials, Fig. 2a). This was then followed by the interleaved session 1, where a negative mismatch was introduced in the experienced audio-tactile sequences by randomly decreasing the whisker stimulus intensity paired with sound A from 30% to 10%, in 15% of the trials (Fig. 2a). In the subsequent interleaved session 2, the negative mismatch was presented in the audio-tactile sequence with sound B (whisker stimulus intensity decreased from 50% to 10%, in 15% of the trials). Once again, we observed that the size of the mismatch response increased with the stimulus deviation, being larger for the Δ40% compared to the Δ20% in stimulus intensity (Fig. 2b, c). Furthermore, the increased size of the mismatch response in the interleaved session 2 also suggests the absence of habituation to the mismatch trials which could result in a decreased mismatch response over time (Supplementary Fig 2a, b). Thus, the PPC can report positive and

negative prediction errors that increase with the deviation from the learned sensory sequence.

## Stimulus variability differentially modulates positive and negative prediction errors

Would the prediction for an association be stronger in an audio-tactile sequence where the whisker stimulus has a fixed intensity as compared to a whisker stimulus with variable intensities? To address this, we now paired the looming sound (sound A) with a whisker stimulus intensity of 50% and the non-looming sound (sound B) with a variable whisker stimulus intensity of 30%, 50% or 70% in a separate imaging session. As before, both audio-tactile sequences were presented randomly to the mice in a pairing session ("pairing" session of 160 to 200 trials, Fig. 3a). In the subsequent interleaved session 1, a positive mismatch was introduced by randomly increasing the whisker stimulus intensity paired with sound A to 90% in 15% of the trials, while the pairing with sound B remained unchanged. In the interleaved session 2 that followed, the positive mismatch was presented in the audio-tactile sequence with sound B (whisker stimulus intensity increased from 30%, 50% or 70% to 90%, in 15% of the trials). Surprisingly, we observed that the positive mismatch generated in the presence of a fixed whisker stimulus intensity in interleaved session 1 was smaller, as compared to the mismatch response associated with a variable whisker stimulus intensity in interleaved

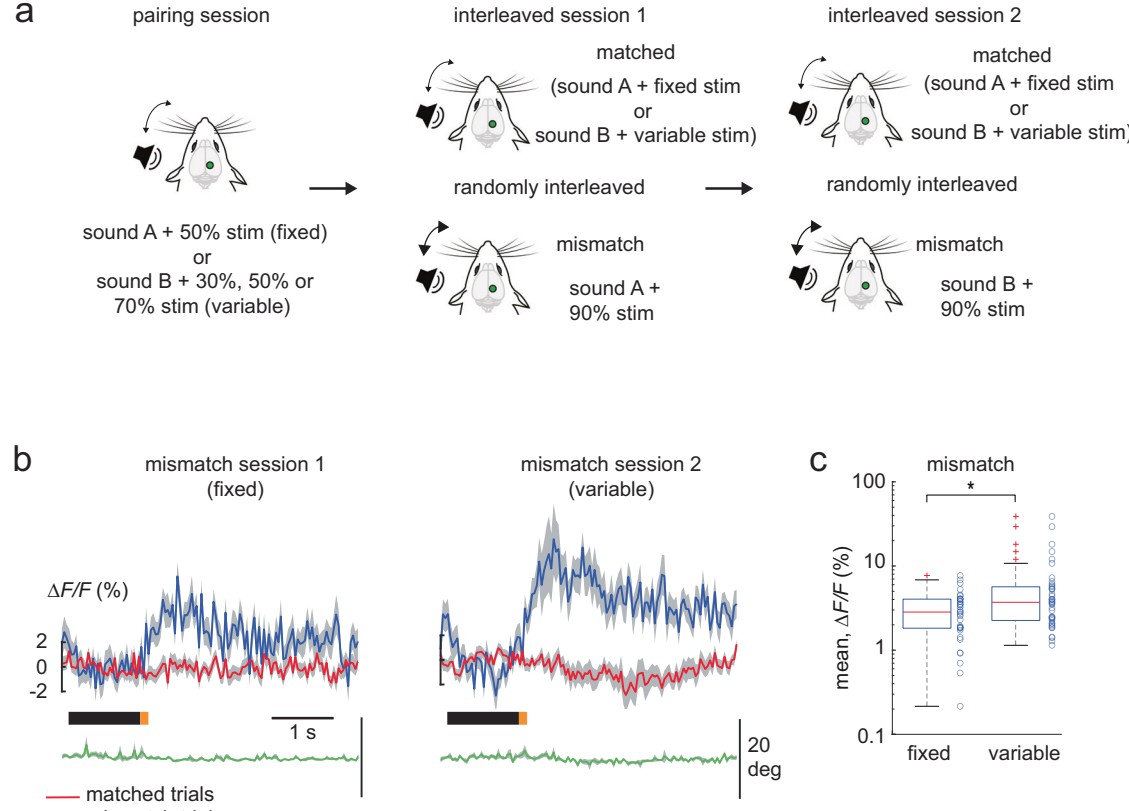

**Fig. 3 | Positive prediction errors increase with whisker stimulus variability.**
**a** Experimental design to evaluate positive prediction errors under stimulus variability in a single imaging session. During the pairing session, sound A is paired with a fixed whisker stimulus intensity of 50%, and sound B with a variable whisker stimulus intensity (30%, 50%, or 70%). Interleaved session 1 and 2 presented positive mismatch trials where the whisker stimulus intensity was randomly increased to 90% in 15% of trials. Hence, the interleaved sessions comprised of randomized matched (sound A with 50% whisker stimulus intensity, sound B with a variable whisker stimulus intensity of 30%, 50% or 70%) and mismatch trials (session 1: sound A with 90% whisker intensity; session 2: sound B with 90% whisker intensity). **b** Population averages of ΔF/F traces of mismatch-responsive neurons in session 1 (fixed whisker stimulus intensity, $n$ = 35 mismatch neurons of 376 neurons) and session 2 (variable

whisker stimulus intensity, $n$ = 42 mismatch neurons of 376 neurons), with their corresponding matched (red) and mismatch (blue) trial averages (4 Thy-1GCaMP6f mice; 6 FOV). **c** Box plot of average population responses of mismatch-responsive neurons, for interleaved sessions 1 (fixed whisker stimulus intensity) and 2 (variable whisker stimulus intensity) as shown in (**b**). mean response ΔF/F: fixed whisker stimulus intensity 3.0 ± 0.3%, variable whisker stimulus intensity 5.9 ± 1.1%. Data are represented as mean ± s.e.m. Boxplot central line indicates the median, the bottom and top edges of the box indicate the 25th and 75th percentiles respectively, the whiskers extend to maximum and minimum points within 1.5 s.d., and outliers are marked with crosses. Statistical significance is indicated by * for $p < 0.05$ with two-sided Wilcoxon-Mann-Whitney test. The mouse illustrations in panel **a** are adapted from ref. 6.

session 2 (Fig. 3b, c). This difference was significant in a late window, 0.65 s post stimulus and not immediately after the stimulus (see Methods). To verify if the nature of the sound (looming vs non-looming) can influence the mismatch responses, we switched the sound pairings between the whisker stimuli in a separate imaging session. That is, we paired sound B (non-looming) with the fixed whisker stimulus and sound A (looming) with the variable whisker stimulus, followed by the interleaved sessions 1 and 2 with the positive mismatch on sound B and sound A, respectively. Again, we found that higher whisker stimulus variability generated a larger positive mismatch response independent of the nature of the sound used in the auditory tactile association (Supplementary Fig. 1) in a late stimulus window. Hence positive prediction errors are larger when the learned association includes stimulus variability, compared to those where the stimulus is fixed.

We then asked if negative prediction errors are comparably modulated by stimulus variability. In a separate imaging session, we presented mice with the same audio-tactile sequence as in the pairing session in Fig. 3a, that is sound A with a fixed whisker stimulus intensity of 50% and sound B with a variable whisker stimulus intensity of 30%, 50% or 70%. This was followed by the interleaved session 1 where a negative mismatch was introduced by randomly decreasing the whisker stimulus intensity paired with sound A from 50% to 10%, in 15% of the trials (Fig. 4a). In the subsequent interleaved

session 2, the negative mismatch was presented with sound B that was paired with the variable whisker stimulus intensity (whisker stimulus intensity decreased from 30%, 50% or 70% to 10%, in 15% of the trials). Contrary to positive prediction errors, the negative prediction error was larger in interleaved session 1 where the whisker stimulus intensity was fixed in the matched audio-tactile sensory sequences, as compared to interleaved session 2 with the variable whisker stimulus matched trials (Fig. 4b, c). Once again, this difference was significant in a later (0.65 s post stimulus) window. These results suggest that positive and negative prediction errors are differentially modulated by stimulus variability in the PPC. We further validated our findings using a mixed-effects model analysis to account for the nested structure of our data (see "Methods"). Similarly with the mixed-effects models, the differences between the mismatch responses were retained, with the exception of Experiment 1, when we compared the differences using all the mismatch neurons (pooled across interleaved session 1 and 2, see Methods).

## Mechanisms underlying the modulation of sensory prediction errors
Our results show an interesting effect of experienced variability on error neuron responses: whereas positive mismatch responses increased, negative mismatch responses decreased with experienced variability. How can this be

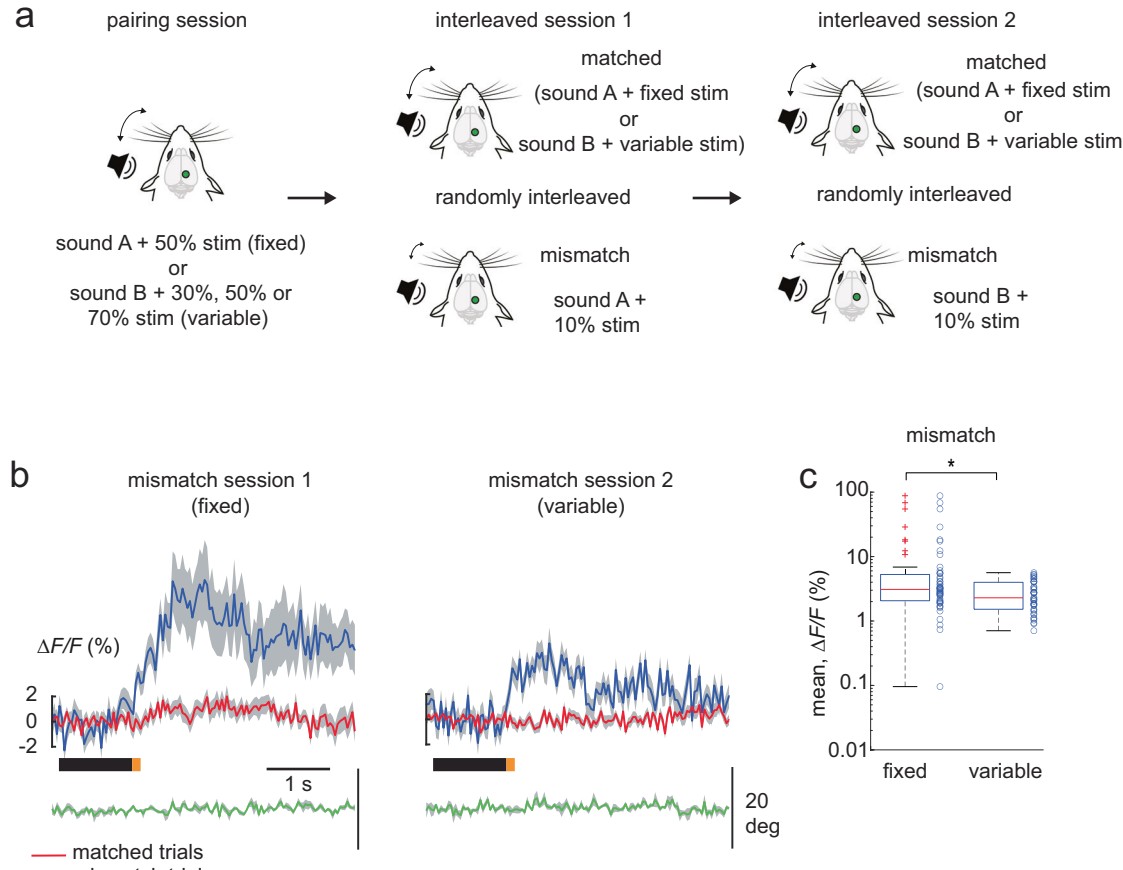

**Fig. 4 | Negative prediction errors decrease with whisker stimulus variability.**
**a** Experimental design to evaluate negative prediction errors under stimulus variability in a single imaging session. Pairing session is the same as that in Fig. 3. Interleaved session 1 and 2 presented negative mismatch trials by randomly decreasing the whisker stimulus intensity to 10% in 15% of trials. Interleaved sessions therefore, included randomized matched (sound A with 50% whisker stimulus intensity, sound B with a variable whisker stimulus intensity of 30%, 50% or 70%) and mismatch trials (session 1: sound A with 10% whisker intensity; session 2: sound B with 10% whisker intensity). **b** Population averages of ΔF/F traces of mismatch-responsive neurons in session 1 (fixed whisker stimulus intensity, $n = 51$ mismatch neurons of 478 neurons) and session 2 (variable whisker stimulus intensity, $n = 42$

mismatch neurons of 478 neurons), with their corresponding matched (red) and mismatch (blue) trial averages (4 Thy-1GCaMP6f mice; 6 FOV). **c** Box plot of average population responses of mismatch-responsive neurons, for interleaved sessions 1 (fixed whisker stimulus intensity) and 2 (variable whisker stimulus intensity) as shown in (**b**). mean response ΔF/F: fixed whisker stimulus intensity 8.4 ± 2.3%, variable whisker stimulus intensity 2.7 ± 0.2. Data are represented as mean ± s.e.m. Boxplot central line indicates the median, the bottom and top edges of the box indicate the 25th and 75th percentiles respectively, the whiskers extend to maximum and minimum points within 1.5 s.d., and outliers are marked with crosses. Statistical significance is indicated by * for $p < 0.05$ with two-sided Wilcoxon-Mann-Whitney test. The mouse illustrations in (**a**) are adapted from ref. 6.

explained? Layer 2/3 error neurons receive both top-down predictive and bottom-up sensory inputs. It is believed that positive error neurons subtract the prediction from the excitatory sensory input via inhibitory interneurons while negative error neurons subtract the sensory input from the prediction (Fig. 5). By modulating the prediction or the sensory inputs or both, which have an opposing influence on the two types of error neurons, the experienced variability could differentially affect the activity of the two types. To understand the mechanism underlying our findings, we formalised all possible combinations of how the variability could influence the prediction E(s) and/or the sensory input s to obtain the observed experimental changes. Four possible models could potentially capture the experimental finding of an increase in the positive and a decrease in the negative errors with variability:

First, the prediction could be modulated by variability. In the *modulated prediction model* (P-model), the variability could scale down the prediction onto both positive and negative error neurons (Fig. 5a), which can be described by

$$pPE = \left\lfloor s - \frac{1}{\sigma^2} E(s) \right\rfloor^+ \tag{1}$$

and

$$nPE = \left\lfloor \frac{1}{\sigma^2} E(s) - s \right\rfloor^+. \tag{2}$$

For higher variability, this results in larger positive errors as the prediction's negative influence is smaller and in smaller negative errors as the prediction's positive influence is smaller.

Second, the stimulus could be modulated by variability. In the *modulated stimulus model* (S-model), the variability could increase the influence of the stimulus on both positive and negative error neurons (Fig. 5b), which can be described by

$$pPE = \lfloor \sigma^2 s - E(s) \rfloor^+ \tag{3}$$

and

$$nPE = \lfloor E(s) - \sigma^2 s \rfloor^+. \tag{4}$$

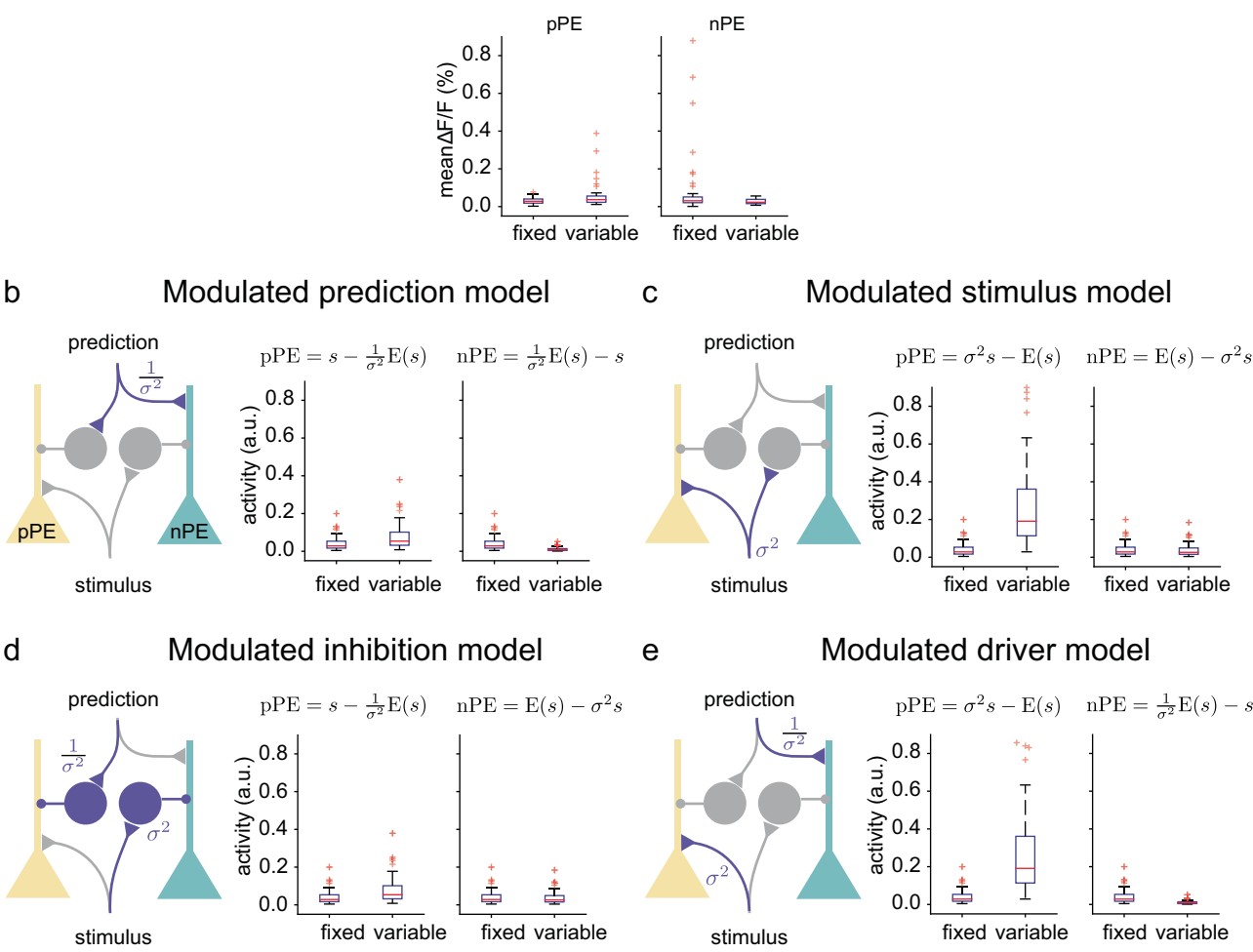

**Fig. 5 | Models of mechanisms underlying the modulation of sensory prediction errors. a** Experimental data as in Figs. 3c, 4c. **b–e** For each panel from left to right: Circuit implementation of the model. Positive prediction error (pPE) neurons subtract the prediction from the stimulus. Negative prediction error (nPE) neurons subtract the stimulus from the prediction. Boxplots of simulated activity of 50 pPE (left) and 50 nPE (right) neurons for the fixed and the variable stimulus condition. **b** Modulated prediction model: the prediction E(s) is inversely modulated by the variability (purple), such that both pPE and nPE neurons receive a smaller prediction for larger variability. **c** Modulated stimulus model: the stimulus s is multiplicatively modulated by the variance, such that both pPE and nPE neurons receive larger stimulus input with larger variability. **d** Modulated inhibition model: the inhibitory terms are modulated by variability. For the pPE the subtracted prediction is inversely modulated by variability. For the nPE, the subtracted stimulus is multiplicatively modulated by the variability. **e** Modulated driver model: the driving inputs to each prediction error neuron are modulated by variability. For the pPE, the excitatory stimulus input is multiplicatively modulated by variability. For the nPE, the excitatory prediction input is inversely modulated by variability. Boxplot central line indicates the median, the bottom and top edges of the box indicate the 25th and 75th percentiles respectively, the whiskers extend to maximum and minimum points within 1.5 s.d., and outliers are marked with crosses.

For higher variability, this results in larger positive errors as the excitation by the stimulus is increased and in smaller negative errors as the inhibition by the stimulus is increased.

Third, the variability could influence the subtraction computation. In the *modulated inhibition model* (I-model), the variability could, on the one hand, scale down the prediction, which is subtracted from positive error neurons and on the other hand, increase the influence of the stimulus, which is subtracted from negative error neurons (Fig. 5c). This model is a mixture of the first two models:

$$pPE = \left\lfloor s - \frac{1}{\sigma^2}\,\mathrm{E}(s) \right\rfloor^+ \tag{5}$$

and

$$nPE = \left\lfloor \mathrm{E}(s) - \sigma^2 s \right\rfloor^+. \tag{6}$$

Finally, the variability could influence the inputs driving the pyramidal cell activity. In the *modulated driver model* (D-model), the variability could, on the one hand, increase the influence of the stimulus onto the positive error neurons and, on the other hand, scale down the prediction onto negative error neurons (Fig. 5d), as described by

$$pPE = \left\lfloor \sigma^2 s - \mathrm{E}(s) \right\rfloor^+ \tag{7}$$

and

$$nPE = \left\lfloor \frac{1}{\sigma^2}\,\mathrm{E}(s) - s \right\rfloor^+. \tag{8}$$

To investigate which models fit the data best, we simulated 50 positive and 50 negative prediction error circuits for each model. To capture the observed variance in error neuron activity, we sampled the synaptic weights in each circuit from a log-normal distribution, as weights are typically

log-normally distributed[11–14] We then compared the resulting model distributions with the data by using a two-sample Kolmogorov-Smirnov test. Both the modulated stimulus model (S-Model, Fig. 5c) and the modulated driver model (D-model, Fig. 5e) predicted very high positive prediction error activity during the variable condition, which we did not observe in the data. The two-sample Kolmogorov-Smirnov test confirmed a significant difference in these cases (Supplementary Fig. 4b, c orange bars). Hence, these two models are less likely to reflect the underlying mechanism. All models yielded very small negative prediction error activity during the variable condition, such that no model could fit this aspect of the data well (Supplementary Fig. 4b). Overall, the modulated inhibition-model (I-model, Supplementary Fig. 4b) provided the best fit. To investigate whether an influence of variability on the magnitude of negative prediction error neuron activity could account for the data, we added an additional multiplicative modulation of negative prediction error neurons by variability. With this addition, the modulated inhibition model (I-model, Fig. 5d) turned out to not be significantly different from the data in any condition (Supplementary Fig. 4c, d), whereas also with this modification, the modulated prediction model (P-Model, Fig. 5b) was still significantly different from the data (Supplementary Fig. 4c, d).

Importantly, the models' validity should ultimately be tested with further experimental investigation. The models make the following distinct experimentally testable predictions: The first two models (P-model and S-model) suggest that the top-down prediction is the same for both positive and negative mismatches. Whereas the P-model implies that the prediction should be inversely modulated by variability, the S-model assumes that variability should not influence the mean sensory prediction error $E(s)$ but the stimulus $s$. The D-model suggests that the prediction should be different between positive and negative error neurons: for negative errors, it should be inversely modulated by variability, whereas for positive errors, it should not be modulated. Finally, in the I-model, there is also a difference between positive and negative errors, but the modulation by variability happens in each circuit on the subtracted variable. In the positive error circuit, the prediction is subtracted via inhibitory interneurons, which may introduce the observed inverse modulation by variability. In the negative error circuit, the stimulus is subtracted by inhibitory interneurons, which may introduce the observed modulation by variability.

## Discussion

In this study, we used an audio-tactile stimulus sequence, where two different sounds were used to predict two distinct whisker stimuli, respectively. We showed that neurons in layer 2/3 of the PPC can report mismatches in these sequences. It has been recently shown that the PPC could report both positive and negative mismatches in the form of sensory prediction errors, as represented by largely non-overlapping populations of neurons[6]. It has been proposed that positive and negative prediction errors are computed by separate classes of neurons[4]. Effectively, the presence of distinct positive and negative prediction error neurons has been shown in V1, where these neurons display opposing responses to visual flow input and mismatch[5].

In V1, mismatch responses have been shown to scale with the speed of locomotion, where the difference between the predicted and actual speed of visual flow is directly correlated to the size of the mismatch response[8,9]. In agreement with this, we have demonstrated that mismatch responses in layer 2/3 neurons of the PPC scale proportionally to the difference between the predicted and actual whisker stimulus. Furthermore, this was observed in both positive and negative prediction error neurons, indicative that in the PPC mismatch processing of audio-tactile sequences can be described within the framework of predictive coding.

We passively exposed mice to auditory tactile sequences which allowed them to predict a fixed or variable stimulus based on different sound contexts in the audio-tactile sequence. As the mice were not required to use this information in the context of goal directed behavior, it is not possible to ascertain if the mice were able to associate the audio tactile sequence.

Nonetheless, we showed that mismatch responses were differentially modulated by stimulus variability. More specifically, we demonstrated that positive prediction errors were larger when the predicted stimulus was variable, and conversely, negative prediction errors were larger when the predicted stimulus was fixed. The difference between different mismatch responses (variable vs fixed) was observed 0.65 s after stimulus onset (see Methods), and not immediately after the stimulus. This suggests the potential role of feedback loops that could enhance these differences. More interestingly, this difference was surprising as Bayesian statistical theory does not differentiate between positive and negative errors and hence predicts that they should be equally modulated by variability[1,15–18] If sensory prediction error responses are used for learning the posterior of the stimulus distribution, an unequal weighting of errors by variability would lead to a biased estimate of the mean of the distribution. Depending on whether the variability in our experiment is interpreted as expected or unexpected uncertainty, we expected either an up- or down-modulation of errors with the variability. Expected or irreducible uncertainty means that the mouse interprets the stimuli as being one stimulus, which is noisy. Then both sensory prediction errors should be down-modulated by uncertainty[19,20]. Alternatively, the variability in our experiment could be interpreted as uncertainty of the environment, i.e. the mouse interprets the stimuli as being different and learns that sudden changes in the stimulus are possible. Then, sensory prediction errors should be upregulated by the variability, as an uncertain environment reduces the reliability of the prior and should lead to a higher weighting of sensory prediction errors[18].

Experimental findings indicate that negative prediction error neurons are excited by the top-down prediction while positive prediction error neurons are inhibited by the prediction[4]. Our findings can hence be potentially described by four models, where variability modulates the influence of the prediction or the stimulus or both on the response of positive and negative prediction error neurons (Fig. 5). The modulated prediction model in which the variability down-modulates the prediction is appealing as also the variability is predicted based on the sound stimulus and hence could be a function of higher-level activity. Although it captures the sign of the changes in the error responses, it is not the model which fits the data best. In the modulated stimulus model, the variability up-regulates the influence of the stimulus. We can only speculate how this happens. One possibility may be that top-down inputs upregulate the activity of stimulus-encoding layer 2/3 pyramidal cells. However, this model predicts a very large increase in the positive prediction error response, which we do not see in the data. The modulated driver model, in which the variability increases the excitatory influence of the stimulus on the positive prediction error neurons and decreases the excitatory influence of the prediction on the negative prediction error neurons, also predicts a very large increase in the positive prediction error, which we do not see in the data. Finally, the modulated inhibition model, in which the inhibitory influence of the prediction on positive prediction errors is down-modulated and the inhibitory influence of the stimulus on the negative prediction error neurons is up-regulated, is appealing as the modulation happens on the inhibitory components of the circuit. Inhibitory neurons are highly modulated by neuromodulators, which could provide a confidence-like signal to the circuit. It also fits the data best. Future experiments are needed to delineate what is happening in the circuit, with the modulated prediction and the modulated inhibition model being the most likely candidates given our data.

Top-down feedback from M2 has been shown to suppress[21], as well as enhance[22–24] sensory processing in primary sensory cortices, as part of a widespread cortical circuit motif[22,25]. Indeed in the PPC, negative prediction errors can be modulated by top-down feedback from M2[6], which can be seen as contributing to the prediction. While we did not record from inhibitory neurons in our experiments, the positive prediction error was smaller in the fixed stimulus session, compared to the session where the variable stimulus was predicted (Fig. 3). Hence top-down feedback from M2 could potentially contribute to part of the prediction that is formalized in all four models that we have proposed.

Overall, our findings show that in the PPC, positive and negative mismatch responses are differentially modulated by stimulus variability, as predicted by an auditory cue. These differences can be potentially exploited in the search for therapeutic approaches for the treatment of mental disorders, where the processing of top-down and bottom-up inputs can be dysregulated.

## Experimental model and subject details

All experimental procedures followed the guidelines of the Veterinary Office of Switzerland and were approved by the Cantonal Veterinary Office in Bern. The data was collected from Thy-1GCaMP6f mice ($n = 5$), and both males and females were used. All mice were at least 8 weeks old at the time of viral injection/head-post implantation.

## Method details

### Experimental design

This study did not involve randomization or blinding. No data or mice were excluded from the analysis.

### Surgery

Mice were anesthetized with isoflurane (3%) and subcutaneously injected with carprofen (5 mg/kg) prior to surgery. For long-term in vivo calcium imaging, a cranial window was implanted over the PPC as described previously[26]. Mice were anesthetized with isoflurane and subcutaneously injected with buprenorphine (0.1 mg/kg) prior to surgery for window implantation. Briefly, a craniotomy was performed over the PPC, at 1.7 mm lateral and 2.0 mm posterior of bregma. A cover glass (4 mm diameter) was placed directly over the exposed dura mater and sealed to the skull with dental acrylic. A metal post was fixed to the skull with dental acrylic, posterior to the cranial window, to allow for subsequent head fixation.

One week after chronic window implantation, mice were handled daily for another week and gradually habituated to head fixation. Intrinsic optical signal (IOS) imaging was performed on the mice to identify the location of PPC by exclusion as previously described[7]. In brief, to avoid the activation of surrounding whiskers, all whiskers except the γ-barrel-column whisker on the right whisker pad of the mice were trimmed and their locations were mapped using IOS on the exposed skull during whisker stimulation (rostrocaudal deflections at 10 Hz). The location of the primary visual cortex (V1) was similarly mapped using full-field stimulation with a green LED placed 5 mm in front of the contralateral eye. The non-activated region between these two identified sites was delineated as PPC for subsequent imaging sessions.

### Presentation of the sensory stimuli sequence and sensory prediction errors scaling in the posterior parietal cortex (Figs. 1 and 2)

To explore how prediction error responses in the PPC scale with the magnitude of sensory deviations, we designed experiments using an audio-tactile sequence. Each mouse was first passively exposed to a pairing session of 160 to 200 trials, in which a specific sound was consistently followed by a whisker stimulus of a defined intensity. In this session (depicted in Fig. 1a), a looming sound (sound A) was paired with a whisker stimulus intensity of 30%, while a non-looming sound (sound B) was associated with a whisker stimulus intensity of 50%. These audio-tactile sequences were randomly presented to awake, head-fixed mice. Immediately after the pairing session, we introduced "interleaved" sessions to determine the impact of positive and negative mismatches in the experienced audio-tactile sequences. These sessions were conducted separately for each type of mismatch and involve distinct imaging sessions. Hence each imaging session consisted of a pairing session and two interleaved sessions (session 1 and 2) which were done sequentially on the same FOV, within an interval of 2 to 3 minutes. In some mice more than one FOV (each FOV was unique) was acquired 4 to 14 days later. There was no overlap of FOVs between the positive and negative mismatch experiments (see Supplementary table 1).

In the first interleaved session (200 trials) in Figs. 1, 2, positive and negative mismatches (in separate sessions) were introduced by randomly adjusting the whisker stimulus intensity paired with sound A from 30% to either 90% or 10% in 15% of the trials. This manipulation allowed us to investigate the effects of deviations from the expected intensity in the audio-tactile sequence. Hence the session comprised randomized matched trials (sound A with whisker stimulus 30%, sound B with whisker stimulus 50%) and mismatch trials (sound A with whisker stimulus 90% or 10%). Subsequently, the second interleaved session was conducted, introducing either a positive or negative mismatch in the audio-tactile sequence with sound B. In this session, the whisker stimulus intensity associated with sound B was adjusted from 50% to 90% or 10% in 15% of the trials.

### Assessment of prediction strength in variable whisker stimulus intensities (Figs. 3 and 4)

In a separate imaging session, a looming sound (sound A) was paired with a fixed whisker stimulus intensity of 50%, while the non-looming sound (sound B) was associated with variable whisker stimulus intensities of 30%, 50%, or 70%. Both audio-tactile sequences were randomly presented to head-fixed mice during a pairing session (see Fig. 3a). Following the pairing session, two interleaved sessions were introduced to evaluate the influence of stimulus variability on positive and negative mismatches in the experienced audio-tactile sequences as described previously. In a subset of control experiments, the non-looming sound (B) was associated with the fixed whisker stimulus instead of the looming sound (A) (see Supplementary Fig. 1).

### Sound and whisker stimuli features

The looming (increasing intensity) and non-looming (constant intensity) sounds were based on previously recorded soundtracks[7] and recreated in MATLAB, that consisted of a cloud of tones (0.1–8 kHz). Each sound was 1 s long and was delivered via a loudspeaker at 75 dB placed contralateral to the imaging site. Tactile stimuli were delivered as deflections ($3 \times 10$ Hz, 1 ms pulse) to multiple whiskers. This was achieved by attaching small metal particles to the whiskers and subsequently moving them via a brief magnetic field generated by a coil placed beneath the head of the mouse. The whisker intensities delivered here (10% to 90% stimulus intensity) were within the range of mice perception, as previously shown in a whisker deflection detection task[27]. Each trial began with a 2 s baseline, followed by a brief sound cue (50 ms, 85 dB) to signal trial start to the mouse. In the pairing and interleaved trials, the looming sound was played 1 s after the sound cue and was followed immediately by the whisker stimulus. Each trial was 8 to 9 s long and trials were presented with an inter-trial interval of 2–5 s, to render them irregular in presentation timing. White noise (65 dB) was played during the entire duration of the imaging session. Sound and whisker stimulus delivery was controlled by custom-written software in C.

### Whisker tracking

The whisker field was illuminated with a 940-nm infrared LED light and movies were acquired at 100 Hz (500 × 500 pixels) using a pixy camera system coupled with an Arduino board to track the position of the whiskers in real time[28].

### Two-photon calcium imaging

We used a custom-built 2-photon microscope controlled by ScanImage 2019 equipped with a Ti:sapphire laser system (~100-fs laser pulses; Mai Tai BB; Newport Spectra Physics), a water-immersion objective (16 × LWDPF, 0.8 NA; Nikon), resonant scan mirrors (model 6210; Cambridge Technology), and a Pockel's Cell (Conoptics) for laser intensity modulation. For calcium imaging, GCaMP6f was excited at 915 nm with the Mai Tai. Emitted fluorescence was collected with red (617/73 nm) and green (520/60 nm) emission filters. Images were acquired at 30 Hz with 512 × 512-pixel resolution.

## Histology

After the calcium imaging recordings, mice were deeply anaesthetized by intraperitoneal injection of 80/10 ketamine/xylazine mixture and transcardially perfused with 4% paraformaldehyde (PFA). Brains were removed and post-fixed in PFA for 24–48 h at 4 °C and were subsequently washed in phosphate-buffered saline and sliced at 100 µm. Brain slices were mounted using Mowiol® 4-88 prior to imaging on a LEICA m205 FCA fluorescence stereo microscope.

## Quantification and statistical analysis

**Two-photon calcium data processing.** Somatic calcium signals were automatically detected using the Python-based CaImAn analysis pipeline, which performed motion correction, source extraction, spike deconvolution and component registration[29]. All detected components were then manually curated with the average and maximum projection images of the acquired data.

Calcium signals are presented here as $\Delta F/F = (F-F_0)/F_0$, where $F_0$ was calculated for each trial as the as the mean of the 1 s window prior to the sound cue that signalled trial start. The responsiveness of a detected neuron to a given stimulus (whisker stimulus or mismatch) was determined by comparing the distribution of their single trial responses (single trial mean $\Delta F/F$ calculated in a 1 s window from stimulus onset minus the mean baseline window 1 s before stimulus onset) against the distribution of 1000 randomly selected events (random baseline corrected mean $\Delta F/F$ calculated in a 1 s window as above). Significance was determined with a two-sided Mann-Whitney-U test ($p < 0.05$). Hence, we were able to identify populations of neurons that were positively or negatively modulated by a given stimulus. We subsequently classified the positively modulated shuffle-corrected neurons as pairing-, matched- or mismatch-responsive, based on the session in which they were significantly responsive. While we did identify neurons that were negatively modulated by the mismatch trials, we did not include them in our analysis as we focused on neurons that were driven by mismatch trials and hence be able to report these mismatch events. We further validated the number of mismatch reporting neurons by determining the 95% confidence interval of a random sample (8–12% of the population, corresponding to the fraction of mismatch neurons we report our experiments) drawn from the shuffled population generated from each of the experiments performed in Figs. 1–4. This allowed us to verify if our detected mismatch responses fall within or outside the true population, as represented by the shuffled population data. For example in Fig. 1, using the shuffled data we obtained the confidence intervals (in $\Delta F/F$ %) of $-1.2 \times 10^{-2}$ to $8.0 \times 10^{-2}$ and $-1.9 \times 10^{-2}$ to $7.2 \times 10^{-2}$ for the interleaved sessions 1 and 2 respectively. The average mismatch responses of all the neurons that we report in Fig. 1 were well outside these confidence intervals, indicating that they are indeed different from the shuffled data and hence not detected by chance. In a comparable way, we obtained confidence intervals of $-1.7 \times 10^{-2}$ to $4.6 \times 10^{-2}$ and $-1.6 \times 10^{-2}$ to $4.7 \times 10^{-2}$ for the interleaved sessions 1 and 2 respectively in Fig. 2, confidence intervals of $-9.1 \times 10^{-2}$ to $3.7 \times 10^{-2}$ and $-8.2 \times 10^{-2}$ to $5.0 \times 10^{-2}$ for the interleaved sessions 1 and 2 respectively in Fig. 3, and confidence intervals of $-0.48$ to $5.5 \times 10^{-2}$ and $-0.53$ to $4.6 \times 10^{-2}$ for the interleaved sessions 1 and 2 respectively in Fig. 4. Population average responses presented here were determined from the averaged traces of each neuron that has been identified as being session responsive in its respective stimulus window. The population average neuronal traces were baseline-corrected by subtracting the mean of the baseline (1 s prior to stimulus onset, corresponding to the sound presentation window) from the entire averaged trace. In Figs. 1 and 2, the mean $\Delta F/F$ is computed during a 1 s window from stimulus onset minus the baseline mean $\Delta F/F$ 1 s prior to stimulus. In Figs. 3, 4 and Sup Fig.1, the mean $\Delta F/F$ is computed during a 1 s window 0.65 s from stimulus onset minus the baseline mean $\Delta F/F$ 1 s prior to stimulus. We did not observe any significant difference between the mismatch responses (fixed vs variable stimulus) when the mean $\Delta F/F$ was computed during the 1 s window from stimulus onset minus the baseline mean $\Delta F/F$ 1 s prior to stimulus, as performed in Figs. 1 and 2. Also comparable late window differences within the positive and negative mismatch responses were not observed in Figs.1 and 2 (variation of the stimulus intensities).

We carried out a mixed-model analysis to verify if the differences between mismatch-responsive neurons can still be observed when nested structure of the data is taken into account. Here the mouse identity was set as the random effect and the interleaved session 1 vs 2 as the fixed effect. While we do not assume a normal distribution of our data, we explored four complementary statistical approaches to ensure robustness. We present in Supplementary Table 2 the results of this analysis using either a standard linear mixed-effects model (assuming normality), a generalized linear mixed-effects model (gamma distribution), a log-transformed linear mixed-effects model or a non-parametric permutation test. Mixed-effects models were implemented using the fitlme and fitglme functions in MATLAB 2025. We show that overall, the differences between mismatch-responsive neurons that we report in Experiments 1 to 4 are still present when mouse identity is taken into account, where some models are better than others in showing these differences (Supplementary Table 2).

When we re-analysed the mismatch responses reported in Experiments 1 to 4 using all mismatch neurons (mismatch-responsive pooled from interleaved session 1 and 2 in each experiment) as a comparison between paired data, we observed no differences between the mismatch responses except in Experiment 3 (Supplementary Fig. 5). This was likely due to mismatch-responsive neurons being largely responsive in their respective mismatch sessions, with smaller or no responses to the other mismatch session (Supplementary Fig. 3). The mixed-effects models however did identify differences between the mismatch responses in Experiments 2, 3 and 4 (Supplementary Table 3). These differences were likely lost in the analysis using pooled data across mice (Supplementary Fig. 5), but preserved in the mixed effects models that account for mouse variability.

## Statistical analysis

All data are presented as mean ± s.e.m. unless stated otherwise. The boxplot central line indicates the median, the bottom and top edges of the box indicate the 25th and 75th percentiles respectively, the whiskers extend to maximum and minimum points within 1.5 s.d., and outliers are marked with crosses. The non-parametric Wilcoxon signed-rank paired test and Wilcoxon-Mann-Whitney test for matched and mismatch group comparisons were performed, respectively. In Fig.1, the comparison of the mismatch responses was done using the non-overlapping neurons, with the Wilcoxon-Mann-Whitney test. The mismatch response of overlapping neurons was done using the Wilcoxon signed-rank test. In Fig.2, the mismatch responses of the non-overlapping neurons were compared with the Wilcoxon-Mann-Whitney test. No comparison of the overlapping neurons as done as there was only 1 neuron. All mismatch responsive neurons (overlapping and non-overlapping) are shown in Figs. 1 and 2, and the means are computed using these neurons. In Figs. 3 and 4, there were no overlapping neurons and the Wilcoxon-Mann-Whitney test was used to compare the mismatch responses. All tests were two-sided. We did not test for a normal distribution of the data.

## Modelling

To model the activities of individual neurons, we simulated 50 positive and 50 negative prediction error circuits, where we sampled the weights from a log-normal distribution with a shape parameter of 1.0. To obtain magnitude of neural responses similar to the data, we normalized the weights such that the maximum weight was 0.05. pPE and nPE are defined according to Eqs. 1–4, where $\sigma^2 = \sigma_0^2 + \sigma_s^2$ with $\sigma_0 = 1$ to avoid division by zero. The prediction, i.e.\ the expected value of the stimulus $E(s) = 5$, the positive mismatch stimulus was 9, and the negative stimulus was 1, to resemble the

experimental setup. The modulated $\sigma_s$ in the fixed condition was 0 and in the variable condition, it was 1.6 (corresponding to the standard deviation of a discrete uniform distribution over the elements [3,5,7]). For the adjusted model with negative prediction error modulation, we additionally multiplied all negative prediction errors by $\sigma_s$. We used Python for the sampling (in particular scipy.lognorm), analysis, and plotting.

## Reporting summary

Further information on research design is available in the Nature Portfolio Reporting Summary linked to this article.

## Data availability

Supplementary data are provided with this paper. The data used to generate the figures and reach the conclusions are available on Zenodo (https://doi.org/10.5281/zenodo.16880929).

## Code availability

The code for the models are available on Zenondo (https://doi.org/10.5281/zenodo.16813620).

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

## Acknowledgements

We thank T. Nevian for resources and critical comments on the manuscript. We thank M. Petrovici for discussions on the model. We thank the members of the Nevian laboratory for comments on the manuscript. We thank C. Dellenbach, C. Käser, E. Scheuner, and J. Burkhalter for their excellent technical support in electronics and mechanics. We thank M. Känzig for his help in animal husbandry.

## Author contributions

B.L., S.K., C.R. and S.S. performed the experiments and analyzed the data. W.S. and K.W. designed the models and formulas. S.K. set up the IOS, whisker tracking, and data analysis pipeline. K.W. and S.S designed the experiments. S.S. built the microscope, secured the funding, and supervised the project. B.L., K.W. and S.S. wrote the manuscript with comments from all authors. This work was supported by the University of Bern and by grants to S.S. from the Swiss National Science Foundation (31003A_182678).

## Competing interests

The authors declare no competing interests.
