## [Transparent Peer Review file · Communications Biology]

Differential modulation of positive and negative prediction errors by stimulus variability in the mouse posterior parietal cortex

Corresponding Author: Dr Shankar Sachidhanandam

Version 0:

Reviewer comments:

Reviewer #2

(Remarks to the Author)

This study reports on four related experiments which were all conducted on the same five mice. The main question is whether neurons in the mouse posterior parietal cortex (PPC) respond differently to identical tactile target stimuli depending on the degree to which their occurrence can be predicted from preceding sound stimuli. In each experiment, two different sounds were first repeatedly paired with whisker vibrations of different intensity. In two subsequent "interleaved" sessions, in 15% of cases one of the two sounds was followed by a stronger tactile "mismatch" stimulus than in the pairing session. If a neuron produced a higher response to the same vibration with different preceding stimuli, it was identified as a positive prediction error (PE) neuron. The other three experiments followed a similar protocol, except that the mismatch stimuli were weaker than expected (Experiments 2, 4) and/or one sound was paired with different intensities during initial pairing (Experiments 3, 4). The conclusion of the study is that the PPC contains both positive and negative PE neurons, and that the response magnitude of PE neurons is affected by the variability of the stimulus used in the pairing sessions. The authors investigate four related neural network models and find that a "modulated inhibition" model provides the best explanation of the data.

Although I find the manuscript interesting and well written, there are some problems with the study design. Also, some of the analyses are inadequate, and I do not agree with all of the authors' conclusions. All issues could be satisfactorily addressed without additional experiments.

Major issues

1) At several instances, the authors imply that the animals have learned to predict the target tactile stimulus (Abstract: "we trained awake head-restrained mice to associate sounds with upcoming tactile stimuli", Results: "we first created an association between...", Discussion: "by training mice to predict a fixed or variable stimulus..."). However, according to Methods the animals were actually passively exposed to stimulus sequence, and no behavioral readouts were obtained. One might argue that the neuronal data suggest that the animals learned to associate sounds with tactile stimuli, but this is not necessarily the case – for example, in humans the P300 oddball response (a type of prediction error) is easily observed without training and even in coma (Neurocrit Care 8: 262), and a mismatch negativity response can be observed in anesthetized untrained mice (Hear Res 408: 108296). Another reason to doubt that animals have formed strong associations is that the same sounds were paired with different tactile stimuli over the course of several consecutive experiments performed on the same mice. In sum, misleading statements about animal training should be deleted from the manuscript, the passive exposure paradigm should be made explicit, and the Discussion should include a caveat that the associative status of the stimuli is not known.

2) Another issue concerns active whisker movements (which were tracked but not analyzed). Neural responses to whisker stimuli are modulated by whisking (e.g., Cerebral Cortex 16:1142). Assuming that the animals have formed an association between sounds and whisker vibrations, they might whisk differently depending on the expected stimulus, which in turn would affect the neural representation of the stimulus (and could serve as an index of associative learning). This is a crucial analysis which could support the authors' conclusions and disentangle bottom-up and top-down processing in this paradigm, thereby adding considerable strength to the manuscript.

3) Another concern relates to the statistical analysis. For example, in the box plots in Figure 1C, there are presumably some neurons that are included in both the delta 60% and in the delta 40% distribution, and some which are included in only one distribution. The Wilcoxon-Mann-Whitney test used by the authors assumes independent samples, but this assumption is violated as some data points are dependent (units included in both distributions).

Relatedly, the PPE and NPE neurons are identified on the basis of a difference in activity in two stimulation conditions. Only 7 to 10% of the total sample is rendered significant, while around 5% would be expected by chance (false positives).

Assuming a binomial distribution, the 95% confidence interval around 5% ranges from 3.4 to 6.6%, so the total number of significant neurons is barely higher than chance expectations. To support their claim that PPE and NPE neurons indeed exist in the PPC, I suggest that the authors assess the true type-1-error rate for their approach (e.g., using a random selection of their 1,000 random samples against the remaining ones), calculate confidence limits on this basis, and then report whether or not the observed number of significant neurons falls outside this confidence interval.

4) The authors claim that the mismatch response does not adapt / habituate (line 135), which is interesting but not well supported by their statement (the MMR might be increased in interleaved session 2 because of the larger delta but still be smaller than if it had occurred in interleaved session 1). A stronger case would be made if the average response magnitude within a single session did not decrease (15 trials, I suppose).

5) Regarding the modeling, arbitrating between the four models is difficult because they all make qualitatively similar predictions – increased PPE and decreased NPE with variable stimuli during pairing. Some models are rejected because they predict “very high PPE activity during variable condition” which was not observed in the data. However, effect sizes for the MI model are also considerably higher than what is seen in the data. In sum, and in relation to my criticism above regarding the identification of PPE and NPE neurons, I think a quantitative analysis of the goodness of fit is required to make a strong case, along with a graphical juxtaposition of model and data points.

Minor issues

1) Please report details on the whisker stimulation (deflection amplitude and velocity). Kinematic information is needed as we do not know whether the mice could actually discriminate the stimuli (e.g., 30% and 50% intensity) if asked to. Have these stimuli been used before in behavioral studies? Or is neuronal data available from tactile areas which shows clearly distinguishable response patterns? Or does the whisker tracking data allow to assess the impact of 30, 50, 90% intensity stimulations?

2) What was the time between consecutive imaging sessions?

3) Line 55 “... we are less able to predict the association between stimuli...” We are not predicting an association but an upcoming stimulus.

4) Line 60: I would add “sensory” before “prediction errors” to clearly distinguish these from reward prediction errors.

5) Line 64: “separate classes of non-overlapping neurons” -> “separate classes of neurons” or “non-overlapping classes of neurons”

6) Line 69f: “encode the variability of the learned sensory associations” – it is not the associations which are variable

7) The box plots in Figure 1C, 2C, 3C, and especially 4C are difficult to assess visually because 80-90% of the y axis range is devoted to a handful of outliers. One way to enlarge the boxes is to render the ordinates logarithmic.

8) In all figures, the meaning of the black and orange bars in the B panels are not defined (I suppose sound and vibration, resp.).

9) The construction of the entire data set is not clear to me. For Experiment 1, the authors report data from five mice, imaged over eight sessions, but each mouse seems to have undergone three sessions (pairing -> interleaved 1 -> interleaved 2).

The figure legend mentions 703 neurons in total for both interleaved sessions, but that would mean at least ten sessions were conducted, assuming mice were not imaged during the pairing session. Or do the authors mean that an imaging session encompasses the entire pairing->interleaved 1&2 conditions, and some animals underwent this more than once? This is suggested by the information there were eight fields of view. Similar for the other experiments.

10) I suggest to use consistent terms for clarity – PE+ or pPE, PE– or nPE (main text and Figure 5).

Reviewer #3

(Remarks to the Author)

1. Brief summary of the manuscript

Leonardon and colleagues investigate how uncertainty influences prediction error responses in individual neurons. Audio-tactile pairs were repeatedly presented to awake mice, with sound serving as a cue for the upcoming tactile stimulus. Using 2-photon calcium imaging, the authors identified neurons in layer 2/3 of the posterior parietal cortex that responded to positive or negative prediction errors (corresponding to stimuli that were stronger or weaker than expected) and scaled their responses with the amplitude of deviation from expectation. Furthermore, they found that negative prediction errors decreased, while positive prediction errors increased with higher variability during the pairing. Modeling of potential circuit mechanisms suggests that computations of both negative and positive prediction errors are mediated by inhibitory neurons.

2. Overall impression of the work

Studying uncertainty in sensory prediction is highly valuable. The experimental design is elegant and well-suited to addressing the research question. The finding of differential modulation in positive and negative prediction errors is both novel and intriguing. The analyses are straightforward and largely support the authors' conclusions. Additionally, the modeling component offers a fresh perspective on potential mechanisms underlying the observed results. While I am highly supportive of the study, I have some concerns that the statistics could be further strengthened or the data presented in

greater detail.

3. Specific comments, with recommendations for addressing each comment

Major comment :

3.1 In Figures 1c, 2c, 3c, and 4c prediction errors are compared between populations of neurons. But many of the data points are not independent because they are sampled from the same few individual animals/recordings. Although there is some level of proof already, the effect could be driven by a single animal/FOV. To strengthen their results, the authors could for instance: (1) compare animal averages, using a Wilcoxon signed rank test, or (2) display/test the effects animal per animal , and discuss how many recordings/animals have it, or if needed (3) replicate the main finding with additional data points.

Minor comments

3.2 - In the models, I find it hard to conceptualize what could be the origin/substrate of the variability's influence on S the sensory input (Fig 5b-d), in particular as presented in Fig 5b and 5d. In part because the effect is sound specific, meaning that upstream sensory neurons must have been already "informed" and modulated with the sound and require specific connectivity for each sound? Only 5c seems plausible, if the variability's influence on S is exerted dynamically (trial by trial) through the intermediate Inhibitory neuron targeting nPE. This logic would argue more for a top down dis-inhibition through these intermediate inhibitory neurons targeting nPE rather than modulation of the sensory inputs. Could the authors additionally discuss the origin/substrate of the variability's influence on S the sensory input ?

3.3 - In the abstract L40, "positive" and "negative" prediction error effects are swapped.

3.4 - I felt that the last abstract sentence (L43) could give a better account for the novelty of model suggestions.

3.5 - Are the same neurons those which respond to mismatch following the two sounds? I think that the analysis of the identity of neurons in the comparison would be valuable as it could further constrain the models if the overlap is weak versus strong.

3.6 - Are only considered the neurons which positively respond to mismatch. Are there also neurons responding negatively, or is it negligible ?

3.7 - Is the proportion of mismatch trials 15% of the total trials or 15% of the corresponding pair (e.g. L101)?

3.8 - The end of the method section "two photon calcium data processing" (L471-474) is confusing. Is the quantification window different in Fig1-2 versus 3-4 ?

3.9 - In fig 1b and subsequent similar figure the bottom bar is not explained (aud. and tactile stimuli I presume). Could it be described in the caption?

3.10 - Are the pairing and test session carried the same day/ in close succession ?

Version 1:

Reviewer comments:

Reviewer #2

(Remarks to the Author)

The authors have satisfactorily addressed my concerns.

Reviewer #3

(Remarks to the Author)

I appreciate the authors' efforts in addressing my previous comments in detail. Below, I provide a few remaining points for consideration, particularly regarding statistical interpretation and data structure

1 - Regarding point 3.1, I appreciate the authors' detailed response and additional analysis regarding overlapping neurons and their use of non-parametric statistics. However, my main concern, i.e. the non-independence due to multiple neurons sampled from the same animals or FOVs, remains insufficiently addressed. The current statistical approach still treats neurons as independent samples, which may inflate statistical significance. I would encourage the authors to either:

- Perform animal-level or FOV-level averaging and re-test the main comparisons using these values as independent units (e.g., via Wilcoxon signed-rank tests across animals), or

- Display and discuss the effects animal-by-animal, to assess robustness and potential outliers, or

- Use a mixed-model analysis with animal or FOV as random effects, and interleaved 1 vs. 2 as a fixed effect

I would also encourage the authors to consider whether the pre-selection of neurons based on prediction-error responses might introduce bias into their comparisons (currently the bias tends to equalize comparison of interleaved 1 vs 2). As an alternative approach, the authors could consider for instance,

- comparing the average response amplitude across all neurons with prediction- error responses (~ pooled neurons from interleaved 1 and 2), for each animal/FOV ;

or - include the same number of neurons in each group , e.g, the 10% top responding in interleaved 1 vs. 2, for each animal or FOV.

Addressing the nested structure of the data (non independence) would greatly strengthen the validity and generalizability of the conclusions from figures 1c, 2c, 3c, and 4c.

2 -All other points have been addressed satisfactorily.

3- I noticed that Supplementary figure 2 and 3 do not currently have a title and caption. it would be appreciable to provide these.

Response to Reviewers' comments

Reviewer #2 (Remarks to the Author):

This study reports on four related experiments which were all conducted on the same five mice. The main question is whether neurons in the mouse posterior parietal cortex (PPC) respond differently to identical tactile target stimuli depending on the degree to which their occurrence can be predicted from preceding sound stimuli. In each experiment, two different sounds were first repeatedly paired with whisker vibrations of different intensity. In two subsequent "interleaved" sessions, in 15% of cases one of the two sounds was followed by a stronger tactile "mismatch" stimulus than in the pairing session. If a neuron produced a higher response to the same vibration with different preceding stimuli, it was identified as a positive prediction error (PE) neuron. The other three experiments followed a similar protocol, except that the mismatch stimuli were weaker than expected (Experiments 2, 4) and/or one sound was paired with different intensities during initial pairing (Experiments 3, 4). The conclusion of the study is that the PPC contains both positive and negative PE neurons, and that the response magnitude of PE neurons is affected by the variability of the stimulus used in the pairing sessions. The authors investigate four related neural network models and find that a "modulated inhibition" model provides the best explanation of the data.

Although I find the manuscript interesting and well written, there are some problems with the study design. Also, some of the analyses are inadequate, and I do not agree with all of the authors' conclusions. All issues could be satisfactorily addressed without additional experiments.

We thank the Reviewer for the critical comments and suggestions, and we address the raised issues below.

Major issues

1) At several instances, the authors imply that the animals have learned to predict the target tactile stimulus (Abstract: "we trained awake head-restrained mice to associate sounds with upcoming tactile stimuli", Results: "we first created an association between...", Discussion: "by training mice to predict a fixed or variable stimulus..."). However, according to Methods the animals were actually passively exposed to stimulus sequence, and no behavioral readouts were obtained. One might argue that the neuronal data suggest that the animals learned to associate sounds with tactile stimuli, but this is not necessarily the case – for example, in humans the P300 oddball response (a type of prediction error) is easily observed without training and even in coma (Neurocrit Care 8: 262), and a mismatch negativity response can be observed in anesthetized untrained mice (Hear Res 408: 108296). Another reason to doubt that animals have formed strong associations is that the same sounds were

paired with different tactile stimuli over the course of several consecutive experiments performed on the same mice. In sum, misleading statements about animal training should be deleted from the manuscript, the passive exposure paradigm should be made explicit, and the Discussion should include a caveat that the associative status of the stimuli is not known.

We thank the Reviewer for this important point. We have removed all statements that might incorrectly suggest that the mice were trained in any way. Furthermore, we now state in the discussion in lines 295, (lines 312 tracked) “We passively exposed mice to auditory tactile sequences which allowed them to predict a fixed or variable stimulus based on different sound contexts in the audio-tactile sequence. As the mice were not required to use this information in the context of goal directed behavior, it is not possible to ascertain if the mice were able to associate the audio tactile sequence. “

2) Another issue concerns active whisker movements (which were tracked but not analyzed). Neural responses to whisker stimuli are modulated by whisking (e.g., Cerebral Cortex 16:1142). Assuming that the animals have formed an association between sounds and whisker vibrations, they might whisk differently depending on the expected stimulus, which in turn would affect the neural representation of the stimulus (and could serve as an index of associative learning). This is a crucial analysis which could support the authors' conclusions and disentangle bottom-up and top-down processing in this paradigm, thereby adding considerable strength to the manuscript.

The Reviewer brings up a valid point that differences in whisking behavior to an expected stimulus could potentially serve as an index of associated learning. We now include the average whisker traces along with the average mismatch responses in the figure panels of Figs1 to 4. for the mismatch trials. We now show that the whisker movements evoked during the mismatch window are comparable between the mismatch types presented and hence cannot account for the differences that we observe.

3) Another concern relates to the statistical analysis. For example, in the box plots in Figure 1C, there are presumably some neurons that are included in both the delta 60% and in the delta 40% distribution, and some which are included in only one distribution. The Wilcoxon-Mann-Whitney test used by the authors assumes independent samples, but this assumption is violated as some data points are dependent (units included in both distributions).

We agree that we were not clear enough in the description of our experimental design. We now include a table in the supplementary information that shows a summary at the mouse level per experiment, in terms of the number of mismatch neurons, the total number of neurons per FOV, the overlapping mismatch neurons in

the interleaved 1 and 2 sessions, and the percentage of mismatch neurons per experiment as a whole. We observed a small number of mismatch neurons (5 out of 114 pPE neurons) in Fig.1. Hence the majority of the mismatch neurons were responsive for a single interleaved session. When we compared the positive mismatch response in the two interleaved sessions using these non-overlapping neurons with the Wilcoxon Mann Whitney test, the mismatch response increased with the stimulus deviation. In Fig.2 where the role of stimulus intensity on negative mismatch responses was evaluated, only 1 neuron overlapped in both interleaved sessions. When we compared the negative mismatch response in the non-overlapping mismatch neurons, the mismatch response once again increased with the stimulus deviation. We observed no overlap in the mismatch responsive neurons in Fig.3 and 4, hence the compared samples were independent. We now provide this information in the statistical analysis section of the Methods. In addition, we now show in Supplementary Fig.3 the average responses of each group of mismatch responding neurons in both interleaved sessions, to further illustrate that the overlap of mismatch responding neurons is very small to none.

Relatedly, the PPE and NPE neurons are identified on the basis of a difference in activity in two stimulation conditions. Only 7 to 10% of the total sample is rendered significant, while around 5% would be expected by chance (false positives). Assuming a binomial distribution, the 95% confidence interval around 5% ranges from 3.4 to 6.6%, so the total number of significant neurons is barely higher than chance expectations. To support their claim that PPE and NPE neurons indeed exist in the PPC, I suggest that the authors assess the true type-1-error rate for their approach (e.g., using a random selection of their 1,000 random samples against the remaining ones), calculate confidence limits on this basis, and then report whether or not the observed number of significant neurons falls outside this confidence interval.

We thank the Reviewer for this very useful suggestion in validating our mismatch responses. We have calculated the 95% confidence interval of a random sample drawn from the shuffled data. Here the shuffled data was generated per figure from the data used to create that figure. The random sample was determined from 8 to 12% of the shuffled population data, a fraction that corresponded to the percentage of mismatch neurons that we have observed in our experiments. We were able to compare the reported single mismatch neuron averages (available in the source data) with the 95% confidence interval of the shuffled population for that particular experiment. We observed that all the mismatch reporting neurons were well outside of the computed confidence intervals, indicating that the mismatch responses that we report are above chance levels that can be due by noise, as represented by the shuffled population data. We now state these confidence intervals in the Methods, from lines 503 (lines 523 tracked).

4) The authors claim that the mismatch response does not adapt / habituate (line 135), which is interesting but not well supported by their statement (the MMR might be increased in interleaved session 2 because of the larger delta but still be smaller

than if it had occurred in interleaved session 1). A stronger case would be made if the average response magnitude within a single session did not decrease (15 trials, I suppose).

We agree with the Reviewer on this point and now show the average mismatch response over trials for each of these experiments in Supplementary Fig.2. These figures show that while the mismatch responses do fluctuate over trials, they show on the whole little adaptation over time. The number of trials per figure is determined by the session with the lowest number of mismatch trials, hence they differ between the graphs. On average there were between 11 to 18 mismatch trials per session.

5) Regarding the modeling, arbitrating between the four models is difficult because they all make qualitatively similar predictions – increased PPE and decreased NPE with variable stimuli during pairing. Some models are rejected because they predict “very high PPE activity during variable condition” which was not observed in the data. However, effect sizes for the MI model are also considerably higher than what is seen in the data. In sum, and in relation to my criticism above regarding the identification of PPE and NPE neurons, I think a quantitative analysis of the goodness of fit is required to make a strong case, along with a graphical juxtaposition of model and data points.

We thank the Reviewer for this important suggestion and now present the modeled data with a goodness of fit to better compare with the experimental results. To get similar effect sizes, we adjusted our model by making the source of noise in our model more biologically realistic. Because the mismatch activity is approximately log-normally distributed, and log-normal distributions of synaptic weights and activities are widely reported, we now sample connection weights in our model from a log-normal distribution. Thereby, the distribution of the activity of all prediction error neurons becomes also lognormally distributed. As a quantitative analysis, we now compare our model distributions with the data distribution with two-sample Kolmogorov-Smirnov tests (Supplementary Fig. 4). We also replotted the data within the same figure to allow for a visual comparison (Fig. 5a).

Minor issues

1) Please report details on the whisker stimulation (deflection amplitude and velocity). Kinematic information is needed as we do not know whether the mice could actually discriminate the stimuli (e.g., 30% and 50% intensity) if asked to. Have these stimuli been used before in behavioral studies? Or is neuronal data available from tactile areas which shows clearly distinguishable response patterns? Or does the whisker tracking data allow to assess the impact of 30, 50, 90% intensity stimulations?

The whisker stimulation intensities that we used have previously been employed and shown to allow mice to perform a whisker deflection detection task. We now state in the Methods lines 449 (lines 469 tracked) “The whisker intensities delivered here

(10% to 90% stimulus intensity) were within the range of mice perception, as previously shown in a whisker deflection detection task”.

2) *What was the time between consecutive imaging sessions?*

We performed the pairing and interleaved sessions (session 1 and 2) sequentially within minutes of each other. In mice where more than 1 FOV was acquired, the imaging between FOVs was done 4 to 14 days later. We now state at line 413 (line 433 tracked) “Hence each imaging session consisted of a pairing session and two interleaved sessions (session 1 and 2) which were done sequentially on the same FOV, within an interval of 2 to 3 minutes. In some mice more than one FOV (each FOV was unique) was acquired 4 to 14 days later. There was no overlap of FOVs between the positive and negative mismatch experiments”.

3) *Line 55 “... we are less able to predict the association between stimuli...” We are not predicting an association but an upcoming stimulus.*

We have now rephrased this in line 56 (line 58 tracked) to state “we are less able to predict the stimuli precisely.”

4) *Line 60: I would add “sensory” before “prediction errors” to clearly distinguish these from reward prediction errors.*

We agree with the Reviewer’s suggestion and have added “sensory” to “prediction errors” throughout the manuscript. We however have left “positive” and “negative prediction errors” as these have been specifically described in sensory prediction errors.

5) *Line 64: “separate classes of non-overlapping neurons” -> “separate classes of neurons” or “non-overlapping classes of neurons”*

We have corrected this as suggested.

6) *Line 69f: “encode the variability of the learned sensory associations” – it is not the associations which are variable*

In line 71 (line 73 tracked) we have now rephrased this to “encode the variability of stimuli within learned sensory associations”.

7) *The box plots in Figure 1C, 2C, 3C, and especially 4C are difficult to assess*

visually because 80-90% of the y axis range is devoted to a handful of outliers. One way to enlarge the boxes is to render the ordinates logarithmic.

As suggested, we now plot the y axis of the box plots in Figs.1c, 2c, 3c and 4c in the logarithmic scale.

8) In all figures, the meaning of the black and orange bars in the B panels are not defined (I suppose sound and vibration, resp.).

We have corrected this omission and now state in the figure legend of Fig.1 that “The black and orange bars represent the sound and whisker stimuli respectively, here and in all the subsequent figures.”

9) The construction of the entire data set is not clear to me. For Experiment 1, the authors report data from five mice, imaged over eight sessions, but each mouse seems to have undergone three sessions (pairing -> interleaved 1 -> interleaved 2). The figure legend mentions 703 neurons in total for both interleaved sessions, but that would mean at least ten sessions were conducted, assuming mice were not imaged during the pairing session. Or do the authors mean that an imaging session encompasses the entire pairing->interleaved 1&2 conditions, and some animals underwent this more than once? This is suggested by the information there were eight fields of view. Similar for the other experiments.

We agree that we were not clear in the description of our experiments. We have clarified this in the manuscript and state at line 413 (line 433 tracked) “Hence each imaging session consisted of a pairing session and two interleaved sessions (session 1 and 2) which were done sequentially on the same FOV, within an interval of 2 to 3 minutes. In some mice more than one FOV (each FOV was unique) was acquired 4 to 14 days later. There was no overlap of FOVs between the positive and negative mismatch experiments”. We also now include additional information of the experiments in Supplemental Information Table 1, which gives a clear description of our acquired data, in terms of the mice used and the number of FOVs from each mouse for each mismatch type studied in the Figs 1 to 4.

10) I suggest to use consistent terms for clarity – PE+ or pPE, PE– or nPE (main text and Figure 5).

We agree with this suggestion and have changed all the terms to pPE and nPE to represent positive prediction errors and negative prediction errors respectively.

Reviewer #3 (Remarks to the Author):

1. Brief summary of the manuscript

Leonardon and colleagues investigate how uncertainty influences prediction error responses in individual neurons. Audio-tactile pairs were repeatedly presented to awake mice, with sound serving as a cue for the upcoming tactile stimulus. Using 2-photon calcium imaging, the authors identified neurons in layer 2/3 of the posterior parietal cortex that responded to positive or negative prediction errors (corresponding to stimuli that were stronger or weaker than expected) and scaled their responses with the amplitude of deviation from expectation. Furthermore, they found that negative prediction errors decreased, while positive prediction errors increased with higher variability during the pairing. Modeling of potential circuit mechanisms suggests that computations of both negative and positive prediction errors are mediated by inhibitory neurons.

2. Overall impression of the work

Studying uncertainty in sensory prediction is highly valuable. The experimental design is elegant and well-suited to addressing the research question. The finding of differential modulation in positive and negative prediction errors is both novel and intriguing. The analyses are straightforward and largely support the authors' conclusions. Additionally, the modeling component offers a fresh perspective on potential mechanisms underlying the observed results. While I am highly supportive of the study, I have some concerns that the statistics could be further strengthened or the data presented in greater detail.

3. Specific comments, with recommendations for addressing each comment

We thank the Reviewer for the critical comments and suggestions, and we address the raised issues below.

Major comment :

3.1 In Figures 1c, 2c, 3c, and 4c prediction errors are compared between populations of neurons. But many of the data points are not independent because they are sampled from the same few individual animals/recordings. Although there is some level of proof already, the effect could be driven by a single animal/FOV. To strengthen their results, the authors could for instance: (1) compare animal averages, using a Wilcoxon signed rank test, or (2) display/test the effects animal per animal , and discuss how many recordings/animals have it, or if needed (3) replicate the main finding with additional data points.

We agree with the Reviewer that the effects we show could potentially be driven by a single FOV. In order to clarify this, we now provide in Supplementary Table 1 a

description of each experiment performed in Figures 1 to 4, in terms of the number of mice used and the number of FOVs in each of these mice. We further describe the total number of neurons and mismatch neurons detected per FOV, along with the overlap of mismatch neurons between the interleaved session 1 and 2 of each mismatch experiment. We show that the overlap between mismatch neurons is very low, and the mismatch responses are significantly different with the Wilcoxon-Mann-Whitney test when performed on the non-overlapping neurons. This is particularly the case in Fig.1, where there is an overlap of 5 mismatch neurons. The mismatch response between these 5 overlapping neurons is similar in size, when compared using the Wilcoxon signed rank test. There was only 1 overlapping neuron in Fig.2, and a comparison of the mismatch responses using the non-overlapping neurons showed a significant difference with the Wilcoxon Mann Whitney test. We did not observe any overlapping mismatch neurons for the experiments in Figs.3 and 4. The comparison of the mismatch responses using these independent samples was significantly different with the Wilcoxon-Mann-Whitney test. We now describe this additional analysis in the Statistical analysis section of the Methods, lines 536 (line 556 tracked) “In Fig.1, the comparison of the mismatch responses was done using the non-overlapping neurons, with the Wilcoxon-Mann-Whitney test. The mismatch response of overlapping neurons was done using the Wilcoxon signed-rank test. In Fig.2, the mismatch responses of the non-overlapping neurons were compared with the Wilcoxon-Mann-Whitney test. No comparison of the overlapping neurons as done as there was only 1 neuron. In Figs. 3 and 4, there were no overlapping neurons and the Wilcoxon-Mann-Whitney test was used to compare the mismatch responses.”

Minor comments

3.2 - In the models, I find it hard to conceptualize what could be the origin/substrate of the variability's influence on S the sensory input (Fig 5b-d), in particular as presented in Fig 5b and 5d. In part because the effect is sound specific, meaning that upstream sensory neurons must have been already “informed” and modulated with the sound and require specific connectivity for each sound? Only 5c seems plausible, if the variability's influence on S is exerted dynamically (trial by trial) through the intermediate Inhibitory neuron targeting nPE. This logic would argue more for a top down dis-inhibition through these intermediate inhibitory neurons targeting nPE rather than modulation of the sensory inputs. Could the authors additionally discuss the origin/substrate of the variability's influence on S the sensory input ?

We thank the Reviewer for this suggestion. We considered all possible models regardless of how plausible they are, because we did not want to judge beforehand. To clarify this, we adjusted our wording in the results section such that it now reads in lines 195 (lines 200 tracked) “To understand the mechanism underlying our findings, we formalised all possible combinations of how the variability could influence the prediction $E(s)$ and/or the sensory input s to obtain the observed experimental

changes. Four possible models could potentially capture the experimental finding of an increase in the positive and a decrease in the negative errors with variability”

We agree that the Inhibition Model is the most plausible and interestingly, it also turns out to be the most likely model, given the data distributions. We now added a sentence at line 333 (line 350 tracked) to the discussion about the origin of the modulation of sensory inputs:” We can only speculate how this happens. One possibility may be that top-down inputs upregulate the activity of stimulus-encoding layer 2/3 pyramidal cells.”

3.3 - In the abstract L40, “positive” and “negative” prediction error effects are swapped.

We thank the Reviewer for pointing out this oversight on our part and we have corrected this.

3.4 - I felt that the last abstract sentence (L43) could give a better account for the novelty of model suggestions.

We thank the Reviewer for this suggestion and have now rephrased this sentence in line 41 (line 42 tracked) to read “Finally, by comparing four computational models, we show that variability-dependent modulation of inhibitory neurons best fits the data, suggesting a previously unrecognized role for inhibition in encoding uncertainty prediction errors”.

3.5 - Are the same neurons those which respond to mismatch following the two sounds? I think that the analysis of the identity of neurons in the comparison would be valuable as it could further constrain the models if the overlap is weak versus strong.

The overlap of neurons that respond to mismatch in the two interleaved sessions is very small, to none. Specifically, we observed 5 neurons that responded to mismatch in interleaved session 1 and 2 in Fig.1 and 1 neuron that responded to mismatch in both interleaved sessions in Fig.2. We did not observe neurons with overlapping mismatch responses in Figs.3 and 4. The model currently treats the mismatch neurons as independent samples as there is no overlap of mismatch neurons in these experiments that examined the role of stimulus variability on mismatch response.

3.6 - Are only considered the neurons which positively respond to mismatch. Are there also neurons responding negatively, or is it negligible ?

Yes, we only took into account neurons that were positively driven by mismatch. This allowed us to match our data to the canonical sensory prediction error circuit (Keller & Mrsic-Flogel, 2017) that addresses such neurons. We did not analyze the neurons that responded negatively to mismatch and include this statement in the methods, line 500 (line 520 tracked) “While we did identify neurons that were negatively modulated by the mismatch trials, we did not include them in our analysis as we focused on neurons that were driven by mismatch trials and hence be able to report these mismatch events.”

3.7 - Is the proportion of mismatch trials 15% of the total trials or 15% of the corresponding pair (e.g. L101)?

The mismatch trials are approximately 15% of the total number of trials presented per interleaved session. This corresponded to about 11 to 18 mismatch trials per interleaved session. The minimum number of mismatch trials per experiment is illustrated in Supplementary Fig.3.

3.8 - The end of the method section “two photon calcium data processing” (L471-474) is confusing. Is the quantification window different in Fig1-2 versus 3-4 ?

It is correct that we used a different quantification window in Figs.1 to 2 compared to Figs 3. and 4. This was done as we observed a difference in the average mismatch response in a later window after the stimulus, and not immediately after. We now clarify this in lines 526 (lines 546 tracked) to state “We did not observe any significant difference between the mismatch responses (fixed vs variable stimulus) when the mean $\Delta F/F$ was computed during the 1s window from stimulus onset minus the baseline mean $\Delta F/F$ 1s prior to stimulus, as performed in Fig. 1 and 2. Also comparable late window differences within the positive and negative mismatch responses were not observed in Fig.1 and 2 (variation of the stimulus intensities).”

3.9 - In fig 1b and subsequent similar figure the bottom bar is not explained (aud. and tactile stimuli I presume). Could it be described in the caption?

We thank the Reviewer for pointing this omission out to us. We now include this description in the figure legend “The black and orange bars represent the sound and whisker stimuli respectively, here and in all the subsequent figures.”

3.10 - Are the pairing and test session carried the same day/ in close succession ?

Yes, it is correct that the pairing and interleaved session 1 and 2 are done on the same day (same imaging session), within minutes of each other. We now state in line 413 (line 433 tracked) “Hence each imaging session consisted of a pairing session

and two interleaved sessions (session 1 and 2) which were done sequentially on the same FOV, within an interval of 2 to 3 minutes.”

Response to Reviewers' comments

Reviewer #3 (Remarks to the Author):

I appreciate the authors' efforts in addressing my previous comments in detail. Below, I provide a few remaining points for consideration, particularly regarding statistical interpretation and data structure

We thank the Reviewer once again for the insightful comments and suggestions and we address the concerns below.

1 - Regarding point 3.1, I appreciate the authors' detailed response and additional analysis regarding overlapping neurons and their use of non-parametric statistics. However, my main concern, i.e. the non-independence due to multiple neurons sampled from the same animals or FOVs, remains insufficiently addressed. The current statistical approach still treats neurons as independent samples, which may inflate statistical significance. I would encourage the authors to either:

- Perform animal-level or FOV-level averaging and re-test the main comparisons using these values as independent units (e.g., via Wilcoxon signed-rank tests across animals), or*
- Display and discuss the effects animal-by-animal, to assess robustness and potential outliers, or*
- Use a mixed-model analysis with animal or FOV as random effects, and interleaved 1 vs. 2 as a fixed effect*

We agree with the Reviewer that data acquired from 2 photon calcium imaging is inherently nested in structure. As suggested, we addressed this using a mixed-effects model analysis with the mouse identity as random effects and the interleaved session 1 vs 2 as the fixed effect. As we do not assume normality in the distribution of our data, we present in Supplementary table 2 the results of this analysis using either a standard linear mixed-effects model (assuming normality), a generalized linear mixed-effects model (gamma distribution), a log-transformed linear mixed-effects model or a non-parametric permutation test. We show that overall, the differences between the mismatch-responsive neurons that we report are still present when mice identity is taken into account. We now state the use of mixed-effects models in the Methods lines 532.

I would also encourage the authors to consider whether the pre-selection of neurons based on prediction-error responses might introduce bias into their comparisons (currently the bias tends to equalize comparison of interleaved 1 vs 2). As an alternative approach, the authors could consider for instance,

- comparing the average response amplitude across all neurons with prediction-error responses (~ pooled neurons from interleaved 1 and 2), for each animal/FOV ;*
- or - include the same number of neurons in each group , e.g, the 10% top responding*

in interleaved 1 vs. 2, for each animal or FOV.

We agree with the Reviewer that indeed our pre-selection and comparison of mismatch responsive neurons (interleaved session 1 vs 2) is biased, as we precisely wanted to compare mismatch responses that are generated under conditions of variability and uncertainty, using the mismatch-responsive neurons. As we illustrate in Supplementary Fig 3., mismatch neurons tend to respond preferentially to their respective mismatch session, with smaller or no responses to the other mismatch session. To strengthen our findings and conclusions, we have repeated the analysis in Experiments 1 to 4 using all mismatch responsive neurons (pooled from interleaved sessions 1 and 2), as suggested by the reviewer. Here we performed a Wilcoxon signed rank test between paired data. We show that such an analysis shows significant differences only in Experiment 3, in Supplementary Fig.5. We further used mixed-effects models (comparable to that in Supplementary table 2) to determine the dependence of our observations on mouse identity, and present the results in Supplementary table 3. We show that apart from Experiment 1, the Experiments 2 to 4 demonstrate differences in mismatch responses that are present across the mice when the mismatch neurons are pooled, in line with the results and conclusions we have shown with session selective mismatch neurons. We now describe this in the Methods line 549.

Addressing the nested structure of the data (non independence) would greatly strengthen the validity and generalizability of the conclusions from figures 1c, 2c, 3c, and 4c.

2 -All other points have been addressed satisfactorily.

3- I noticed that Supplementary figure 2 and 3 do not currently have a title and caption. it would be appreciable to provide these.

We have rectified this shortcoming and now present the titles and captions within these figures, in addition to the main text.